# Assessing Disparate Impact of Personalized Interventions: Identifiability and Bounds

**Nathan Kallus**
Cornell University
New York, NY
kallus@cornell.edu

**Angela Zhou**
Cornell University
New York, NY
az434@cornell.edu

## Abstract

Personalized interventions in social services, education, and healthcare leverage individual-level causal effect predictions in order to give the best treatment to each individual or to prioritize program interventions for the individuals most likely to benefit. While the sensitivity of these domains compels us to evaluate the fairness of such policies, we show that actually auditing their disparate impacts per standard observational metrics, such as true positive rates, is impossible since ground truths are unknown. Whether our data is experimental or observational, an individual's actual outcome under an intervention different than that received can never be known, only predicted based on features. We prove how we can nonetheless point-identify these quantities under the additional assumption of monotone treatment response, which may be reasonable in many applications. We further provide a sensitivity analysis for this assumption by means of sharp partial-identification bounds under violations of monotonicity of varying strengths. We show how to use our results to audit personalized interventions using partially-identified ROC and xROC curves and demonstrate this in a case study of a French job training dataset.

## 1 Introduction

The expanding use of predictive algorithms in the public sector for risk assessment has sparked recent concern and study of fairness considerations [3, 9, 10]. One critique of the use of predictive risk assessment argues that the discussion should be reframed to instead focus on the role of *positive interventions* in distributing beneficial resources, such as directing pre-trial services to prevent recidivism, rather than in meting out pre-trial detention based on a risk prediction [8]; or using risk assessment in child welfare services to provide families with additional childcare resources rather than to inform the allocation of harmful suspicion [29, 62]. However, due to limited resources, interventions are necessarily targeted. Recent research specifically investigates the use of models that predict an intervention's benefit in order to efficiently target their allocation, such as in developing triage tools to target homeless youth [46, 57]. Both ethics and law compel such personalized interventions to be fair and to avoid disparities in how they impact different groups defined by certain protected attributes, such as race, age, or gender.

The delivery of interventions to better target those individuals deemed most likely to respond well, even if a prediction or policy allocation rule does not have access to the protected attribute, might still result in disparate impact (with regards to social welfare) for the same reasons that these disparities occur in machine learning classification models [21]. (See Appendix C for an expanded discussion on our use of the term "disparate impact.") However, in the problem of personalized interventions, the "fundamental problem of causal inference," that outcomes are not observed for interventions not administered, poses a fundamental challenge for evaluating the fairness of any intervention allocation rule, as the true "labels" of intervention efficacy of any individual are never observed in the dataset. Metrics commonly assessed in the study of fairness in machine learning, such as group true positive

and false positive rates, are therefore conditional on potential outcomes which are not observed in the data and therefore cannot be computed as in standard classification problems.

The problem of personalized policy learning has surfaced in econometrics and computer science [13, 36, 37, 37, 45, 51], gaining renewed attention alongside recent advances in causal inference and machine learning [4, 14, 28, 63]. In particular, [17] analyze optimal treatment allocations for malaria bednets with nonparametric plug-in estimates of conditional average treatment effects, accounting for budget restrictions; [27] use the generalized random forests method of [64] to evaluate heterogeneity of causal effects in a program matching at-risk youth in Chicago with summer jobs on outcomes and crime; and [46] use BART [32] to analyze heterogeneity of treatment effect for allocation of homeless youth to different interventions, remarking that studying fairness considerations for algorithmically-guided interventions is necessary.

In this paper, we address the challenges of assessing the disparate impact of such personalized intervention rules in the face of unknown ground truth labels. We show that we can actually obtain point identification of common observational fairness metrics under the assumption of *monotone treatment response*. We motivate this assumption and discuss why it might be natural in settings where interventions only either help or do nothing. Recognizing nonetheless that this assumption is not actually testable, we show how to conduct sensitivity analyses for fairness metrics. In particular, we show how to obtain sharp partial identification bounds on the metrics of interest as we vary the strength of violation of the assumption. We then show to use these tools to visualize disparities using partially identified ROC and xROC curves. We illustrate all of this in a case study of personalized job training based on a dataset from a French field experiment.

## 2  Problem Setup

We suppose we have data on individuals $(X, A, T, Y)$ consisting of:

- Prognostic features $X \in \mathcal{X}$, upon which interventions are personalized;
- Sensitive attribute $A \in \mathcal{A}$, against which disparate impact will be measured;
- Binary treatment indicator $T \in \{0, 1\}$, indicating intervention exposure; and
- Binary response outcome $Y \in \{0, 1\}$, indicating the benefit to the individual.

Our convention is to identify $T = 1$ with an active intervention, such as job training or a homeless prevention program, and $T = 0$ with lack thereof. Similarly, we assume that a positive outcome, $Y = 1$, is associated with a beneficial event for the individual, e.g., successful employment or non-recidivation. Using the Neyman-Rubin potential outcome framework [34], we let $Y(0), Y(1) \in \{0, 1\}$ denote the potential outcomes of each treatment. We let the observed outcome be the potential outcome of the assigned treatment, $Y = Y(T)$, encapsulating non-interference and consistency assumptions, also known as SUTVA [60]. Importantly, for any one individual, we never *simultaneously* observe $Y(0)$ and $Y(1)$. This is sometimes termed the fundamental problem of causal inference. We assume our data either came from a randomized controlled trial (the most common case) or an unconfounded observational study so that the treatment assignment is ignorable, that is, $Y(1), Y(0) \perp\!\!\!\perp T \mid X, A$.

When both treatment and potential outcomes are binary, we can exhaustively enumerate the four possible realizations of potential outcomes as $(Y(0), Y(1)) \in \{0, 1\}^2$. We call units with $(Y(0), Y(1)) = (0, 1)$ responders, $(Y(0), Y(1)) = (1, 0)$ anti-responders, and $Y(0) = Y(1)$ non-responders. Such a decomposition is also common in instrumental variable analysis [2] where the binary outcome is take-up of treatment with the analogous nomenclature of compliers, never-takers, always-takers, and defiers. In the context of talking about an actual outcome, following [52], we replace this nomenclature with the notion of response rather than compliance. We remind the reader that due to the fundamental problem of causal inference, response type is *unobserved*.

We denote the conditional probabilities of each response type by

$$p_{ij} = p_{ij}(X, A) = \mathbb{P}(Y(0) = i, Y(1) = j \mid X, A).$$

By exhaustiveness of these types, $p_{00} + p_{01} + p_{10} + p_{11} = 1$. (Note $p_{ij}$ are *random variables*.)

We consider evaluating the fairness of a personalized intervention policy $Z = Z(X, A) \in \{0, 1\}$, which assigns interventions based on observable features $X, A$ (potentially just $X$). Note that by definition, the intervention has zero effect on non-responders, negative effect on anti-responders, and a positive effect only on responders. Therefore, in seeking to benefit individuals with limited

resources, the personalized intervention policy should seek to target only the responders. Naturally, response type is unobserved and the policy can only mete out interventions based on observables.

In classification settings, minimum-error classifiers on the efficient frontier of type-I and -II errors are given by Bayes classifiers that threshold the probability of a positive label. In personalized interventions, policies that are on the efficient frontier of social welfare (fraction of positive outcomes, $\mathbb{P}(Y(Z) = 1)$) and program cost (fraction intervened on, $\mathbb{P}(Z = 1)$) are given by thresholding ($Z = \mathbb{I}[\tau \geq \theta]$) the *conditional average treatment effect* (CATE):

$$
\begin{aligned}
\tau = \tau(X, A) &= \mathbb{E}[Y(1) - Y(0) \mid X, A] = p_{01} - p_{10} \\
&= \mathbb{P}(Y = 1 \mid T = 1, X, A) - \mathbb{P}(Y = 1 \mid T = 0, X, A),
\end{aligned}
$$

where the latter equality follows by the assumed ignorable treatment assignment. Estimating $\tau$ from unconfounded data using flexible models has been the subject of much recent work [32, 61, 64].

We consider observational fairness metrics in analogy to the classification setting, where the "true label" of an individual is their *responder status*, $R = \mathbb{I}[Y(1) > Y(0)]$. We define the analogous true positive rate and true negative rate for the intervention assignment $Z$, conditional on the (unobserved) events of an individual being a responder or non-responder, respectively:

$$
\text{TPR}_a = \mathbb{P}(Z = 1 \mid A = a, Y(1) > Y(0)), \quad \text{TNR}_a = \mathbb{P}(Z = 0 \mid A = a, Y(1) \leq Y(0)). \quad (1)
$$

## 2.1 Interpreting Disparities for Personalized Interventions

The use of predictive models to deliver interventions can induce disparate impact if responding (respectively, non-responding) individuals of different groups receive the intervention at dispropor­tionate rates under the treatment policy. This can occur even with efficient policies that threshold the true CATE $\tau$ and can arise from the disparate predictiveness of $X, A$ of response type (i.e., how far $p_{ij}$ are from 0 and 1). This is problematic because the choice of features $X$ is usually made by the intervening agent (e.g., government agency, etc.).

We discuss one possible interpretation of TPR or TNR disparities in this setting when the intervention is the bestowal of a benefit, like access to job training or case management. From the point of view of the intervening agent, there are specific program goals, such as employment of the target individual within 6 months. Therefore, false positives are costly due to program cost and false negatives are missed opportunities. But outcomes also affect the individual's utility. Discrepancies in TPR across values of $A$ are of concern since they suggest that the needs of those who could actually benefit from intervention (responders) in one group are not being met at the same rates as in other groups. Arguably, for benefit-bestowing interventions, TPR discrepancies are of greater concern. Nonetheless, from the point of view of the individual, the intervention may always grant some positive resource (e.g., from the point of view of well-being), regardless of responder status, since it corresponds to access to a good (and the individual can gain other benefits from job training that may not necessarily align with the intervener's program goals, such as employment in 1 year or personal enrichment). If so, then TNR discrepancies across values of $A$ imply a "disparate benefit of the doubt" such that the policy disparately over-benefits one group over another using the limited public resource without the cover of advancing the public program's goal, which may raise fairness and envy concerns, especially since this "waste" is at the cost of more slots for responders.

Beyond assessing disparities in TPR and TNR for one fixed policy, we will also use our ability to assess these over varying CATE thresholds in order to compute xAUC metrics [41] in Section 6. These give the disparity between the probabilities that a non-responder from group $a$ is ranked above a responder from group $b$ and vice-versa. Thus, they measure the disproportionate access one group gets relative to another in *any* allocation of resources that is non-decreasing in CATE.

We emphasize that the identification arguments and bounds that we present on fairness metrics are primarily intended to facilitate the *assessment* of disparities, which may require further inquiry as to their morality and legality, not necessarily to promote statistical parity via adjustments such as group-specific thresholds, though that is also possible using our tools. We defer a more detailed discussion to Section 8 and re-emphasize that assessing the distribution of outcome-conditional model errors are of central importance both in machine learning [10, 30, 55] and in the economic efficiency of targeting resources [16, 18, 54].

# 3  Related Work

[50] consider estimating joint treatment effects of race and treatment under a deep latent variable model to reconstruct unobserved confounding. For evaluating fairness of policies derived from estimated effects, they consider the gap in population accuracy $\mathrm{Acc}_a = \mathbb{P}\left(Z = Z^* \mid A = a\right)$, where $Z^* = \mathbb{I}[\tau(X) > 0]$ is the (identifiable) optimal policy. In contrast, we highlight the unfairness of even optimal policies and focus on outcome-conditional error rates (TPR, TNR), where the non-identifiability of responder status introduces challenges regarding identifiability.

The issue of model evaluation under the censoring problem of selective labels has been discussed in situations such as pretrial detention, where detention censors outcomes [40, 48]. Sensitivity analysis to account for possible unmeasured confounders is used in [35, 39]. The distinction is that we focus on the targeted delivery of interventions with unknown (but estimated) causal effects, rather than considering classifications that induce one-sided censoring but have definitionally known effects. Recently, partial identification approaches has also been proposed in the case of known outcomes but missing protected attributes [22, 42].

Our emphasis is distinct from other work discussing fairness and causality that uses graphical causal models to decompose predictive models along causal pathways and assessing the normative validity of path-specific effects [44, 47], such as the effect of probabilistic hypothetical interventions on race variables or other potentially immutable protected attributes. When discussing treatments, we here consider interventions corresponding to allocation of concrete resources (e.g., give job training), which are in fact physically manipulable by an intervening agent. The correlation of the intervention's *conditional average* treatment effects by, say, race and its implications for downstream resource allocation are our primary concern.

There is extensive literature on partial identification, e.g. [53], including for individual-level causal effect [43]. In contrast to previous work that analyzes partial identification of average treatment effects when data is confounded and using monotonicity to improve precision [6, 15, 53], we focus on unconfounded (e.g., RCT) data and achieve full identification by assuming monotonicity and consider sensitivity analysis bounds for *nonlinear* functionals of partially identified sets, namely, true positive and false positive rates.

# 4  Identifiability of Disparate Impact Metrics

Since the definitions of the disparate impact metrics in Eq. (1) are conditioned on an unobserved event, such as the response event $Y(1) > Y(0)$, they actually cannot be identified from the data, even under ignorable treatment. That is, the values of $\mathrm{TPR}_a, \mathrm{TNR}_a$ can vary even when the joint distribution of $(X, A, T, Y)$ remains the same, meaning the data we see cannot possibly tell us about the specific value of $\mathrm{TPR}_a, \mathrm{TNR}_a$.

**Proposition 1.** $\mathrm{TPR}_a, \mathrm{TNR}_a$ *(or discrepancies therein over groups) are generally not identifiable.*

Essentially, Proposition 1 follows because the data only identifies the marginals $p_{10} + p_{11}$, $p_{01} + p_{11}$ while $\mathrm{TPR}_a, \mathrm{TNR}_a$ depend on the joint via $p_{01}$, which can vary even while marginals are fixed. Since this can vary independently across values of $A$, discrepancies are not identifiable either.

## 4.1  Identification under monotonicity

We next show identifiability if we impose the additional assumption of monotone treatment response.

**Assumption 1** (Monotone treatment response). $Y(1) \geq Y(0)$. (Equivalently, $p_{10} = 0$.)

Assumption 1 says that anti-responders do not exist. In other words, the treatment either does nothing (e.g., an individual would have gotten a job or not gotten a job, regardless of receiving job training) or it benefits the individual (would get a job if and only if receive job training), but it never harms the individual. This assumption is reasonable for positive interventions. As [38] points out, policy learning in this setting is equivalent to the binary classification problem of predicting responder status.

**Proposition 2.** *Under Assumption 1,*

$$
\begin{aligned}
\mathrm{TPR}_a &= \frac{\mathbb{E}\left[\tau \mid A = a, Z = 1\right] \mathbb{P}\left(Z = 1 \mid A = a\right)}{\mathbb{E}\left[\tau \mid A = a\right]}, \\
\mathrm{TNR}_a &= \frac{\mathbb{E}\left[(1 - \tau) \mid A = a, Z = 0\right] \mathbb{P}\left(Z = 0 \mid A = a\right)}{\mathbb{E}\left[(1 - \tau) \mid A = a\right]}.
\end{aligned}
\tag{2}
$$

Since the quantities on the right hand sides in Eq. (2) are in terms of identified quantities (functions of the distribution of $(X, A, T, Y)$), this proves identifiability. Given a sample and an estimate of $\tau$, it also provides a simple recipe for estimation by replacing each average or probability by a sample version, since both $A$ and $Z$ are discrete. More generally, since these averages are average treatment effects (over subpopulations defined by $A, Z$ values), these quantities can also alternatively be estimated by any average treatment effect estimator and plugged in. For example, we can use doubly robust estimators to ensure specification-robustness [58] or double ML estimators to ensure efficiency when $X$ may be high-dimensional [23].

Thus, Proposition 2 provides a novel means of assessing disparate impact of personalized interventions under monotone response. This is relevant because monotonicity is a defensible assumption in the case of many interventions that bestow an additional benefit, good, or resource, such as the ones mentioned in Section 1. Nonetheless, the validity of Assumption 1 is itself not identifiable. Therefore, should it fail even slightly, it is not immediately clear whether these disparity estimates can be relied upon. We therefore next study a sensitivity analysis by means of constructing partial identification bounds for $\mathrm{TPR}_a, \mathrm{TNR}_a$.

## 5 Partial Identification Bounds for Sensitivity Analysis

We next study the partial identification of disparate impact metrics when Assumption 1 fails, i.e., $p_{10} \neq 0$. We first state a more general version of Proposition 2. For any $\eta = \eta(X, A)$, let

$$\rho_a^{\mathrm{TPR}}(\eta) \coloneqq \frac{\mathbb{E}\left[\tau + \eta \mid A = a, Z = 1\right] \mathbb{P}\left(Z = 1 \mid A = a\right)}{\mathbb{E}\left[\tau + \eta \mid A = a\right]},$$

$$\rho_a^{\mathrm{TNR}}(\eta) \coloneqq \frac{\mathbb{E}\left[1 - (\tau + \eta) \mid A = a, Z = 0\right] \mathbb{P}\left(Z = 0 \mid A = a\right)}{\mathbb{E}\left[1 - (\tau + \eta) \mid A = a\right]}.$$

**Proposition 3.** $\mathrm{TPR}_a = \rho_a^{\mathrm{TPR}}(p_{10})$, $\mathrm{TNR}_a = \rho_a^{\mathrm{TNR}}(p_{10})$.

Since the anti-responder probability $p_{10}$ is unknown, we cannot use Proposition 3 to identify $\mathrm{TPR}_a, \mathrm{TNR}_a$. We instead use Proposition 3 to compute bounds on them by restricting $p_{10}$ to be in an uncertainty set. Formally, given an uncertainty set $\mathcal{U}$ for $p_{10}$ (i.e., a set of functions of $x, a$), we define the simultaneous identification region of the TPR and TNR for all groups $a \in \mathcal{A}$ as:

$$\Theta = \{\left(\rho_a^{\mathrm{TPR}}(\eta), \rho_a^{\mathrm{TNR}}(\eta)\right)_{a \in \mathcal{A}} : \eta \in \mathcal{U}\} \subseteq \mathbb{R}^{2 \times |\mathcal{A}|}.$$

For brevity, we will let $\rho_a(\eta) = \left(\rho_a^{\mathrm{TPR}}(\eta), \rho_a^{\mathrm{TNR}}(\eta)\right)$ and $\rho(\eta) = (\rho_a(\eta))_{a \in \mathcal{A}}$.

The set $\Theta$ describes all possible simultaneous values of the group-conditional true positive and true negative rates. As long as $\forall \eta \in \mathcal{U}$ we have $0 \leq \eta(X, A) \leq \min\left(\mathbb{P}\left(Y = 1 \mid T = 0, X, A\right), \mathbb{P}\left(Y = 0 \mid T = 1, X, A\right)\right)$ (which is identified from the data) by Proposition 3 this set is necessarily sharp [53] given only the restriction that $p_{10} \in \mathcal{U}$. (In particular, this bound on $\eta$ can be achieved by just point-wise clipping $\mathcal{U}$ with this identifiable bound as necessary.) That is, given a joint on $(X, A, T, Y)$, on the one hand, every $\rho \in \Theta$ is realized by some full joint distribution on $(X, A, T, Y(0), Y(1))$ with $p_{10} \in \mathcal{U}$, and on the other hand, every such joint gives rise to a $\rho \in \Theta$. In other words, $\Theta$ is an *exact* characterization of the in-fact possible simultaneous values of the group-conditional TPRs and TNRs.

Therefore, if, for example, we are interested in the minimal and maximal possible values for the true (unknown) TPR discrepancy between groups $a$ and $b$, we should seek to compute $\inf_{\rho \in \Theta} \rho_a^{\mathrm{TPR}} - \rho_b^{\mathrm{TPR}}$ and $\sup_{\rho \in \Theta} \rho_a^{\mathrm{TPR}} - \rho_b^{\mathrm{TPR}}$. More generally, for any $\mu \in \mathbb{R}^{2 \times |\mathcal{A}|}$, we may wish to compute

$$h_\Theta(\mu) \coloneqq \sup_{\rho \in \Theta} \mu^\top \rho. \tag{3}$$

Note that this, for example, covers the above example since for any $\mu$ we can also take $-\mu$. The function $h_\Theta$ is known as the *support function* of $\Theta$ [59]. Not only does the support function provide the maximal and minimal contrasts in a set, it also exactly characterizes its convex hull. That is, $\mathrm{Conv}\left(\Theta\right) = \left\{\rho : \mu^\top \rho \leq h_\Theta(\mu) \,\forall \mu\right\}$. So computing $h_\Theta$ allows us to compute $\mathrm{Conv}\left(\Theta\right)$.

Our next result gives an explicit program to compute the support function when $\mathcal{U}$ has a product form of within-group uncertainty sets:

$$\mathcal{U} = \{\eta : \eta(\,\cdot\,, a) \in \mathcal{U}_a \,\forall a \in \mathcal{A}\}, \tag{4}$$

which leads to $\Theta = \prod_{a \in \mathcal{A}} \Theta_a$ where $\Theta_a = \{\rho_a(\eta_a) : \eta_a \in \mathcal{U}_a\}$.

**Proposition 4.** *Let $r_a^z \coloneqq \mathbb{P}\left(Z = z \mid A = a\right)$ and $\tau_a^z \coloneqq \mathbb{E}\left[\tau \mid A = a, Z = z\right]$. Suppose $\mathcal{U}$ is as in* (4). *Then Eq.* (3) *can be reformulated as:*

$$h_\Theta(\mu) = \sum_{a \in \mathcal{A}} h_{\Theta_a}(\mu_a), \quad \text{where}$$

$$h_{\Theta_a}(\mu_a) = \sup_{\omega_a, t_a} \quad \mu_a^{\text{TPR}} r_a^1 \left(t_a \tau_a^1 + \mathbb{E}\left[\omega_a(X) \mid A = a, Z = 1\right]\right)$$

$$+ \frac{\mu_a^{\text{TNR}} r_a^0}{t_a - 1} (t_a \left(1 - \tau_a^0\right) + \mathbb{E}\left[\omega_a(X) \mid A = a, Z = 0\right])$$

$$\text{s.t.} \quad \omega_a(\cdot) \in t_a \, \mathcal{U}_a, \ t_a \left(r_a^0 \tau_a^0 + r_a^1 \tau_a^1\right) + \mathbb{E}\left[\omega_a \mid A = a\right] = 1.$$

For a fixed value of $t_a$, the above program is a linear program, given that $\mathcal{U}_a$ is linearly representable. Therefore a solution may be found by grid search on the univariate $t_a$. Moreover, if $\mu_a^{\text{TPR}} = 0$ or $\mu_a^{\text{TNR}} = 0$, the above remains a linear program even with $t_a$ as a variable [20]. With this, we are able to express group-level disparities through assessing the support function at specific contrast vectors $\mu$.

### 5.1 Partial Identification under Relaxed Monotone Treatment Response

We next consider the implications of the above for the following relaxation of the monotone treatment response assumption:

**Assumption 2** (*B*-relaxed monotone treatment response). $p_{10} \leq B$.

Note that Assumption 2 with $B = 0$ recovers Assumption 1 and Assumption 2 with $B = 1$ is a vacuous assumption. In between these two extremes we can consider milder or stronger violations of monotone response and the partial identification bounds they corresponds to. This provides us with a means of sensitivity analysis of the disparities we measure, recognizing that monotone response may not hold exactly and that disparities may not be exactly identifiable. For the rest of the paper, we focus solely on partial identification under Assumption 2. Note that Assumption 2 corresponds exactly to the uncertainty set $\mathcal{U}_B = \{\eta : 0 \leq \eta(X, A) \leq \min\left(B, \mathbb{P}\left(Y = 1 \mid T = 0, X, A\right), \mathbb{P}\left(Y = 0 \mid T = 1, X, A\right)\right)\}$. We define $\Theta_B = \prod_{a \in \mathcal{A}} \Theta_{B,a}$ to be the corresponding identification region.

Under Assumption 2, our bounds take on a particularly simple form. Let $\mathcal{B}_a^z(B) = \mathbb{E}\left[\min\left(B, \mathbb{P}\left(Y = 1 \mid T = 0, X, A\right), \mathbb{P}\left(Y = 0 \mid T = 1, X, A\right)\right) \mid A = a, Z = z\right]$ and define

$$\overline{\rho}_a^{\text{TPR}}(B) = \frac{(\tau_a^1 + \mathcal{B}_a^1(B))r_a^1}{\tau_a^0 r_a^0 + (\tau_a^1 + \mathcal{B}_a^1(B))r_a^1}, \quad \overline{\rho}_a^{\text{TNR}}(B) = \frac{(1 - \tau_a^0)r_a^0}{(1 - \tau_a^0)r_a^0 + (1 - \tau_a^1 - \mathcal{B}_a^1(B))r_a^1},$$

$$\underline{\rho}_a^{\text{TPR}}(B) = \frac{\tau_a^1 r_a^1}{(\tau_a^0 + \mathcal{B}_a^0(B))r_a^0 + \tau_a^1 r_a^1}, \quad \underline{\rho}_a^{\text{TNR}}(B) = \frac{(1 - \tau_a^0 - \mathcal{B}_a^0(B))r_a^0}{(1 - \tau_a^0 - \mathcal{B}_a^0(B))r_a^0 + (1 - \tau_a^1)r_a^1}.$$

**Proposition 5.** *Suppose Assumption 2 holds. Then $[\underline{\rho}_a^{\text{TPR}}(B), \overline{\rho}_a^{\text{TPR}}(B)]$ and $[\underline{\rho}_a^{\text{TNR}}(B), \overline{\rho}_a^{\text{TNR}}(B)]$ are the sharp identification intervals for $\text{TPR}_a$ and $\text{TNR}_a$, respectively. Moreover, $(\underline{\rho}_a^{\text{TPR}}(B), \underline{\rho}_a^{\text{TNR}}(B)) \in \Theta_{B,a}$ and $(\overline{\rho}_a^{\text{TPR}}(B), \overline{\rho}_a^{\text{TNR}}(B)) \in \Theta_{B,a}$, i.e., the two extremes are simultaneously achievable.*

## 6 Partial Identification of Group Disparities and ROC and xROC Curves

We discuss diagnostics to summarize possible impact disparities across a range of possible policies.

**TPR and TNR disparity.** Discrepancies in model errors (TPR or TNR) are of interest when auditing classification performance on different groups with a given, fixed policy $Z$. Under Assumption 1, they are identified by Proposition 2. Under violations of Assumption 1, we can consider their partial identification bounds. If the *minimal* disparity remains nonzero, that provides strong evidence of disparity. Similarly, if the *maximal* disparity is large, a responsible decision maker should be concerned about the possibility of a disparity.

Under Assumption 2, Proposition 5 provides that the sharp identification intervals of $\text{TPR}_a - \text{TPR}_b$ and $\text{TNR}_a - \text{TNR}_b$ are, respectively, given by

$$[\underline{\rho}_a^{\text{TPR}}(B) - \overline{\rho}_b^{\text{TPR}}(B), \ \overline{\rho}_a^{\text{TPR}}(B) - \underline{\rho}_b^{\text{TPR}}(B)],$$
$$[\underline{\rho}_a^{\text{TNR}}(B) - \overline{\rho}_b^{\text{TNR}}(B), \ \overline{\rho}_a^{\text{TNR}}(B) - \underline{\rho}_b^{\text{TNR}}(B)].$$
(5)

Given effect scores $\tau$, we can then use this to plot *disparity curves* by plotting the endpoints of Eq. (5) for policies $Z = \mathbb{I}[\tau \geq \theta]$ for varying thresholds $\theta$.

**Robust ROC Curves**   We first define the analogous *group-conditional ROC curve* corresponding to a CATE function $\tau$. These are the parametric curves traced out by the pairs $(1 - \mathrm{TNR}_a, \mathrm{TPR}_a)$ of policies that threshold the CATE for varying thresholds. To make explicit that we are now computing metrics for different policies, we use the notation $\rho(\eta; \tau \geq \theta)$ to refer to the metrics of the policy $Z = \mathbb{I}[\tau \geq \theta]$. Under Assumption 1, Proposition 2 provides point identification of the group-conditional ROC curve:

$$\mathrm{ROC}_a(\tau) := \{(1 - \rho_a^{\mathrm{TNR}}(0; \tau \geq \theta), \rho_a^{\mathrm{TPR}}(0; \tau \geq \theta)) : \theta \in \mathbb{R}\}$$

When Assumption 1 fails, we cannot point identify $\mathrm{TPR}_a, \mathrm{TNR}_a$ and correspondingly we cannot identify $\mathrm{ROC}_a(\tau)$. We instead define the *robust ROC* curve as the union of all partially identified ROC curves. Specifically:

$$\Theta_a^{\mathrm{ROC}}(\tau) := \{(1 - \rho_a^{\mathrm{TNR}}(\eta_a; \tau \geq \theta), \rho_a^{\mathrm{TPR}}(\eta_a; \tau \geq \theta)) : \theta \in \mathbb{R}, \eta_a \in \mathcal{U}_a\}.$$

Plotted, this set provides a visual representation of the region that the true ROC curve can lie in. We next prove that under Assumption 2, we can easily compute this set as the area between two curves.

**Proposition 6.** *Let $\mathcal{U} = \mathcal{U}_B$. Then $\Theta_a^{\mathrm{ROC}}(\tau)$ is given as the area between the two parametric curves $\underline{\mathrm{ROC}}_a(\tau) := \{(1 - \underline{\rho}_a^{\mathrm{TNR}}(B; \tau \geq \theta), \underline{\rho}_a^{\mathrm{TPR}}(B; \tau \geq \theta)) : \theta \in \mathbb{R}\}$ and $\overline{\mathrm{ROC}}_a(\tau) := \{(1 - \overline{\rho}_a^{\mathrm{TNR}}(B; \tau \geq \theta), \overline{\rho}_a^{\mathrm{TPR}}(B; \tau \geq \theta)) : \theta \in \mathbb{R}\}$.*

This follows because the extremes are simultaneously achievable as noted in Proposition 5. We highlight, however, that the lower (resp., upper) ROC curve may not be simultaneously realizable.

**Robust xROC Curves**   Comparison of group-conditional ROC curves may not necessarily show impact disparities as, even in standard classification settings ROC curves can overlap despite disparate impacts [30, 41]. At the same time, comparing disparities for fixed policies $Z$ with fixed thresholds may not accurately capture the impact of using $\tau$ for rankings. [41] develop the xAUC metric for assessing the *bipartite ranking* quality of risk scores, as well as the analogous notion of a xROC curve which parametrically plots the TPR of one group vs. the FPR of *another group*, at any fixed threshold. This is relevant if effect scores $\tau$ are used for downstream decisions by different facilities with different budget constraints or if the score is intended to be used by a "human-in-the-loop" exercising additional judgment, e.g., individual caseworkers as in the encouragement design of [12].

Under Assumption 1, we can point identify $\mathrm{TPR}_a, \mathrm{TNR}_a$, so, following [41], we can define the point-identified xROC curve as

$$\mathrm{xROC}_{a,b}(\tau) = \{(1 - \rho_b^{\mathrm{TNR}}(0; \tau \geq \theta), \rho_a^{\mathrm{TPR}}(0; \tau \geq \theta)) : \theta \in \mathbb{R}\}.$$

Without Assumption 1, we analogously define the *robust xROC* curve as the union of all partially identified xROC curves:

$$\Theta_{a,b}^{\mathrm{xROC}}(\tau) = \{(1 - \rho_b^{\mathrm{TNR}}(\eta_a; \tau \geq \theta), \rho_a^{\mathrm{TPR}}(\eta_a; \tau \geq \theta)) : \theta \in \mathbb{R}, \eta_a \in \mathcal{U}_a\}.$$

**Proposition 7.** *Let $\mathcal{U} = \mathcal{U}_B$. Then $\Theta_{a,b}^{\mathrm{xROC}}(\tau)$ is given as the area between the two parametric curves $\underline{\mathrm{xROC}}_{a,b}(\tau) := \{(1 - \underline{\rho}_b^{\mathrm{TNR}}(B; \tau \geq \theta), \underline{\rho}_a^{\mathrm{TPR}}(B; \tau \geq \theta)) : \theta \in \mathbb{R}\}$ and $\overline{\mathrm{xROC}}_{a,b}(\tau) := \{(1 - \overline{\rho}_b^{\mathrm{TNR}}(B; \tau \geq \theta), \overline{\rho}_a^{\mathrm{TPR}}(B; \tau \geq \theta)) : \theta \in \mathbb{R}\}.$*

This follows because $\mathcal{U}_B$ takes the form of a product set over $a \in \mathcal{A}$.

## 7   Case Study: Personalized Job Training (Behaghel et al.)

We consider a case study from a three-armed large randomized controlled trial that randomly assigned job-seekers in France to a control-group, a job training program managed by a public vendor, and an out-sourced program managed by a private vendor [11]. While the original experiment was interested in the design of contracts for program service delivery, we consider a task of heterogeneous causal effect estimation, motivated by interest in personalizing different types of counseling or active labor market programs that would be beneficial for the individual. Recent work in policy learning has also considered personalized job training assignment [45, 63] and suggested excluding sensitive attributes from the input to the decision rule for fairness considerations, but without consideration of fairness in the causal effect estimation itself and how significant impact disparities may still remain after excising sensitive attributes because of it.

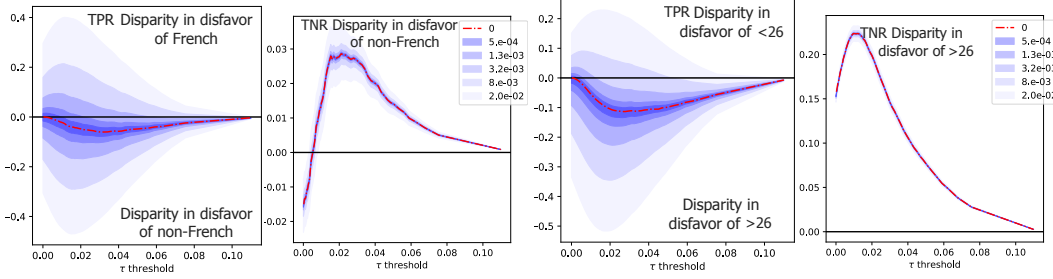

Figure 1: TPR and TNR disparity curves and bounds on French job training dataset (Eq. (5))

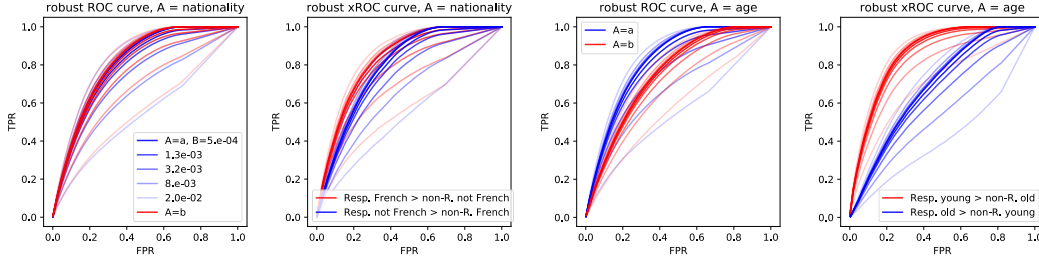

Figure 2: ROC and xROC for $A$ = nationality, age on French job training dataset

We focus on the public program vs. control arm, which enrolled about 7950 participants in total, with $n_1 = 3385$ participants in the public program. The treatment arm, $T = 1$, corresponds to assignment to the public program. The original analysis suggests a small but statistically significant positive treatment effect of the public program, with an ATE of 0.023. We omit further details on the data processing to Appendix B. We consider the group indicators: *nationality* (0, 1 denoting French nationals vs. non-French, respectively), *gender* (denoting woman vs. non-woman), and *age* (below the age of 26 vs. above). (Figures for gender appear in Appendix B.)

In Fig. 1, we plot the identified "disparity curves" of Eq. (5) corresponding to the maximal and minimal sensitivity bounds on TPR and TNR disparity between groups. Levels of shading correspond to different values of $B$, with color legend at right. We learn $\tau$ by the Generalized Random Forests method of [5, 64] and use sample splitting, learning $\tau$ on half the data and using our methods to assess bounds on $\rho^{\mathrm{TPR}}, \rho^{\mathrm{TNR}}$ and other quantities with out-of-sample estimates on the other half of the data. We bootstrap over 50 sampled splits and average disparity curves to reduce sample uncertainty.

In general, the small probability of being a responder leads to increased sensitivity of TPR estimates (wide identification bands). The curves and sensitivity bounds suggest that with respect to nationality and gender, there is small or no disparity in true positive rates but the true negative rates for nationality, gender, and age may differ significantly across groups, such that non-women would have a higher chance of being bestowed job-training benefits when they are in fact not responders. However, TPR disparity by age appears to hold with as much as -0.1 difference, with older actually-responding individuals being less likely to be given job training than younger individuals. Overall, this suggests that differences in heterogeneous treatment effects across age categories could lead to significant adverse impact on older individuals.

This is similarly reflected in the robust ROC, xROC curves (Fig. 2). Despite possibly small differences in ROCs, the xROCs indicate strong disparities: the sensitivity analysis suggests that the likelihood of ranking a non-responding young individual above a responding old individual (xAUC [41]) is clearly larger than the symmetric error, meaning that older individuals who benefit from the treatment may be disproportionately shut out of it as seats are instead given to non-responding younger individuals.

## 8 Discussion and Conclusion

We presented identification results and bounds for assessing disparate model errors of causal-effect maximizing treatment policies, which can lead disparities in access to those who stand to benefit from treatment across groups. Whether this is "unfair" would naturally rely on one's normative

assumptions. One such is "claims across outcomes," that individuals have a claim to the public intervention if they stand to benefit, which can be understood within [1]'s axiomatic justification of fair distribution. There may also be other justice-based considerations, e.g. minimax fairness. We discuss this more extensively in Appendix C.

With the new ability to *assess* disparities using our results, a second natural question is whether these disparities warrant adjustment, which is easy to do given our tools combined with the approach of [30]. This question again is dependent both on one's viewpoint and ultimately on the problem context, and we discuss it further in Appendix C. Regardless of normative viewpoints, auditing allocative disparities that would arise from the implementation of a personalized rule must be a crucial step of a responsible and convincing program evaluation. We presented fundamental identification limits to such assessments but provided sensitivity analyses that can support reliable auditing.

## Acknowledgements

This material is based upon work supported by the National Science Foundation under Grant No. 1846210. This research was funded in part by JPMorgan Chase & Co. Any views or opinions expressed herein are solely those of the authors listed, and may differ from the views and opinions expressed by JPMorgan Chase & Co. or its affiliates. This material is not a product of the Research Department of J.P. Morgan Securities LLC. This material should not be construed as an individual recommendation for any particular client and is not intended as a recommendation of particular securities, financial instruments or strategies for a particular client. This material does not constitute a solicitation or offer in any jurisdiction.

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
