[Supplementary Material · neurips_2019_supp.pdf]



# A Omitted proofs

*Proof of Proposition 1.* To prove this we exhibit a simple example satisfying ignorability where both $\mathrm{TPR}_a, \mathrm{TNR}_a$ and differences therein varies while the joint distribution of $(X, A, T, Y)$ does not.

Let $\mathcal{X} = \{0, 1\}$, $Z = \mathbb{I}[X = 1]$, $\mathbb{P}(T = t, X = x \mid A = a) = \frac{1}{4}$, $\mathbb{P}(A = a) = 1/|\mathcal{A}|$. To specify a joint distribution of $(X, A, T, Z, Y(1), Y(0))$ that satisfies ignorable treatment, it only remains to specify $p_{ij}$.

Note that in this case

$$\mathrm{TPR}_a = \frac{p_{01}(1, a)}{p_{01}(0, a) + p_{01}(1, a)}, \quad \mathrm{TPR}_a = \frac{1 - p_{01}(0, a)}{2 - p_{01}(0, a) - p_{01}(1, a)}.$$

The result follows by noting that where the corresponding joint distribution of $(X, A, T, Z, Y)$ is completely specified by $p_{01} + p_{11}$, $p_{10} + p_{11}$, while $p_{01}$ could vary as long as these sums are neither 0 nor 1. Since we can vary this independently across values of $A$, differences are not identifiable either. $\square$

*Proof of Proposition 2.*

$$\mathbb{P}(Z = 1 \mid A = a, Y(1) > Y(0))$$
$$= \frac{\mathbb{P}(Y(1) > Y(0) \mid A = a, Z = 1)\mathbb{P}(Z = 1 \mid A = a)}{\mathbb{P}(Y(1) > Y(0) \mid A = a)}$$
$$= \frac{\mathbb{E}[\mathbb{E}[Y(1) = 1 \mid {}^{T=1}_{A=a, X=x}] - \mathbb{E}[Y(0) = 1 \mid {}^{T=0}_{A=a, X=x}] \mid {}^{Z=1}_{A=a}]\mathbb{P}(Z = 1 \mid A = a)}{\mathbb{E}[\mathbb{P}(Y(1) = 1 \mid {}^{T=1}_{A=a, X=x}) - \mathbb{P}(Y(0) = 1 \mid {}^{T=0}_{A=a, X=x}) \mid A = a]}$$
$$= \frac{\mathbb{E}[\tau(X, A) \mid {}^{Z=1}_{A=a}]\mathbb{P}(Z = 1 \mid A = a)}{\mathbb{E}[\tau(X, A) \mid A = a]}$$

where the first equality holds by Bayes' rule, the second by iterating expectations on $X$ and Assumption 1, and the third by unconfoundedness and consistency of potential outcomes. The proof for identification of TNR is identical for the quantity $\mathbb{P}(Z = 0 \mid A = a, Y(1) \leq Y(0))$. $\square$

*Proof of Proposition 3.* Recalling that CATE identifies, under violations of Assumption 1

$$\tau = \mathbb{E}[Y(1) - Y(0) \mid X, A] = p_{01} - p_{10},$$

$$= \frac{\mathbb{E}[\tau + \eta \mid {}^{Z=1}_{A=a}]\mathbb{P}(Z = 1 \mid A = a)}{\mathbb{E}[\tau + \eta \mid A = a]} = \frac{(p_{01} - p_{10} + p_{10})\mathbb{P}(Z = 1 \mid A = a)}{\mathbb{E}[(p_{01} - p_{10} + p_{10}) \mid A = a]}$$
$$= \mathbb{P}(Z = 1 \mid A = a, Y(1) > Y(0))$$

$\square$

*Proof of Proposition 4.* The support function evaluated at $\mu$ is:

$$\max_\eta \sum_{a \in \mathcal{A}} \mu_a^{\mathrm{TPR}} \frac{\mathbb{E}[\tau + \eta \mid A = a, Z = 1] r_a^1}{\mathbb{E}[\tau + \eta \mid A = a]} + \mu_a^{\mathrm{TNR}} \frac{\mathbb{E}[1 - (\tau + \eta) \mid A = a, Z = 0] r_0^a}{\mathbb{E}[1 - (\tau + \eta) \mid A = a]}$$

s.t. $0 \leq \eta(x, a) \leq \min(B, \mathbb{P}(Y = 1 \mid T = 0, X, A), \mathbb{P}(Y = 0 \mid T = 1, X, A)), \quad \forall x \in \mathcal{X}, \, \forall a \in \mathcal{A}$

We apply the Charnes-Cooper transformation [18] with the bijection $t_a = \frac{1}{\mathbb{E}[\tau + \eta \mid A = a]}$, $\omega_a = \eta t_a$. The denominator of the second term under this bijection is equivalently

$$\mathbb{E}[1 - (\tau + \eta) \mid A = a] = 1 - \frac{1}{t_a}$$

such that we can rewrite the second term as

$$\mu_a^{\mathrm{TNR}} r_a^0 \left( \frac{1}{1 - 1/t_a} \mathbb{E}[1 - \tau \mid {}^{A=a}_{Z=0}] + \frac{1/t_a}{1 - 1/t_a} \mathbb{E}[\omega_a \mid {}^{A=a}_{Z=0}] \right) = \frac{\mu_a^{\mathrm{TNR}} r_a^0}{t_a - 1} (t \, \mathbb{E}[1 - \tau \mid {}^{A=a}_{Z=0}] + \mathbb{E}[\omega_a \mid {}^{A=a}_{Z=0}])$$

and the objective function overall as:

$$\max_{\eta} \sum_{a \in \mathcal{A}} (\mu_a^{\mathrm{TPR}} r_a^1)(t_a \tau_a^1 + \mathbb{E}\left[\omega_a \mid {}^{A=a}_{Z=1}\right]) + \frac{\mu_a^{\mathrm{TNR}} r_a^0}{t_a - 1}(t_a \left(1 - \tau_a^0\right) + \mathbb{E}\left[\omega_a \mid {}^{A=a}_{Z=0}\right])$$

The new constraint set (including the constraint yielding the definition of $t_a$) is:

$$\{\mathbb{E}\left[\tau t_a + \omega_a \mid A = a\right] = 1, \omega_a \in \mathcal{U}_a\}$$

□

*Proof of Proposition 5.* We first consider the case of maximizing or minimizing the TPR.

We leverage the invariance in the objective function under the surjection on $\eta(x, a)$ to its marginal expectation over a $Z = z, A = a$ partition.

$$w(x, a) = \begin{cases} \mathbb{E}\left[\eta \mid Z = 1, A = a\right] & \text{if } Z = 1 \\ \mathbb{E}\left[\eta \mid Z = 0, A = a\right] & \text{if } Z = 0 \end{cases}$$

Therefore we can reparametrize the program as optimizing over coefficients $x, y$ of the optimal solution, $w^*(x, y) = x\mathbb{I}[Z = 0, A = a] + y\mathbb{I}[Z = 1, A = a]$, with $x \leq \mathcal{B}_a^0(B), y \leq \mathcal{B}_a^1(B)$. Define the fractional objective

$$\begin{aligned} g^*(x, y) &= \frac{\mathbb{E}\left[\tau + x\mathbb{I}[Z = 0] + y\mathbb{I}[Z = 1] \mid A = a, Z = 1\right] \mathbb{P}\left(Z = 1 \mid A = a\right)}{\mathbb{E}\left[\tau + x\mathbb{I}[Z = 0] + y\mathbb{I}[Z = 1] \mid A = a\right]} \\ &= \frac{(\mathbb{E}\left[\tau \mid A = a, Z = 1\right] + y)r_a^1}{\mathbb{E}\left[\tau \mid A = a\right] + xr_a^0 + yr_a^1} \end{aligned}$$

First note that without loss of generality that when maximizing, we can set $x = 0$ since this decreases the objective regardless of the value of $y$. We can consider the constrained problem $\max_{y \leq \mathcal{B}_a^1(B)} h(y)$ where $h(y) = g(0, y)$. Then we have the first and second derivatives,

$$\frac{\partial h}{\partial y} = \frac{r_a^1(\mathbb{E}\left[\tau \mid A = a\right] - \mathbb{E}\left[\tau \mid Z = 1, A = a\right]}{(yr_a^1 + \mathbb{E}\left[\tau \mid A = a\right])^2}, \frac{\partial^2 h}{\partial y^2} = \frac{(r_a^1)^2(\mathbb{E}\left[\tau \mid A = a\right] - \mathbb{E}\left[\tau \mid Z = 1, A = a\right])}{(yr_a^1 + \mathbb{E}\left[\tau \mid A = a\right])^3}$$

By inspection, since $y \geq 0$ we have that $\frac{\partial^2 h}{\partial y^2} \geq 0$ so the function is convex. So when maximizing $h$ on the constraints for $y$, it attains optimal value at the boundary (since $h$ is increasing). When minimizing, note that the derivative is not vanishing anywhere on the constraint set so it suffices to check the endpoints, where the minimum is achieved at $y = 0$.

We now consider the case of minimizing or maximizing the TNR.

We again leverage the symmetry of the solution and reparametrize the program as optimizing over coefficients $x, y$ of the optimal solution, $w^*(x, y) = x\mathbb{I}[Z = 0, A = a] + y\mathbb{I}[Z = 1, A = a]$, with $x \leq \mathcal{B}_a^0(B), y \leq \mathcal{B}_a^1(B)$. Now consider a generic $f(x) = \frac{a - bx}{c - bx - dy}$ which represents the TNR sensitivity bound with $\omega = x\mathbb{I}[Z = 0] + y\mathbb{I}[Z = 1]$, and the constants

$$\begin{aligned} a &= r_a^0 - \mathbb{E}[\tau \mid Z = 0, A = a], \quad c = 1 - \mathbb{E}[\tau \mid A = a] \\ b &= r_a^0, \quad d = r_a^1 \end{aligned}$$

Without loss of generality we know that we can set $y$ to its upper bound $\mathcal{B}_a^1(B)$ when maximizing as we are only increasing the objective value; then $c' = c - \mathcal{B}_a^1(B)r_a^1$. We verify that the second derivative is negative, so that the function is concave:

$$\frac{\partial^2 f}{\partial x^2} = \frac{2b^2(a - c')}{(c' - bx)^3} = \frac{2(r_a^0)^2(r_a^0 - \mathbb{E}[\tau \mid Z = 0, A = a] - (1 - \mathbb{E}[\tau \mid A = a] - \mathcal{B}_a^1(B)r_a^1))}{(1 - \mathbb{E}[\tau \mid A = a] - \mathcal{B}_a^1(B)r_a^1) - xr_a^0}$$

Checking the sign of the numerator simplifies to checking the sign of

$$a - c' = (-r_a^1 + \mathbb{E}[\tau + \mathcal{B}_a^1(B) \mid Z = 1, A = a]))$$

488 which is negative. The denominator is lower bounded by $1 - \mathbb{E}[\tau \mid A = a] - \mathcal{B}_a^1(B)$ which is
489 always positive: therefore the problem is concave. The first derivative $\frac{\partial f}{\partial x} = \frac{b(a-c')}{(c'-bx)^2}$ is negative
490 on the domain; therefore the maximum is achieved at $x = 0$. Therefore, when maximizing, $\omega = $
491 $\mathcal{B}_a^1(B)\mathbb{I}[Z = 1]$.

492 For minimizing the TNR, we take a similar approach: analogously, we can set $y$ to its lower bound
493 without loss of generality. Following the same analysis, the function is still concave $\frac{\partial^2 f}{\partial x^2} = \frac{2b^2(a-c')}{(c'-bx)^3}$
494 since $-r_a^1 - \mathbb{E}[\tau \mid A = a] < 0$ and decreasing with nonzero first-derivative; so the minimum is
495 achieved at $\omega = \mathbb{I}[Z = 0]\mathcal{B}_a^1(B)$.

496 $\qquad\qquad\qquad\qquad\qquad\qquad\qquad\qquad\qquad\qquad\qquad\qquad\qquad\qquad\qquad\qquad\qquad\quad\square$

# B  Behaghel et al. Job Training

498 Reproducibility discussion:

499 - Data preprocessing and exclusion: We processed the data using replication files available
500   with the AEJ: Applied Economics journal electronic supplement. For the sake of simplicity,
501   we analyze the trial as if it were a randomized controlled trial (without accounting for
502   noncompliance or different randomization probabilities that differ by region). Thus, we
503   consider intention-to-treat effects (as intention to treat is ultimately the policy lever available).
504   We further restricted some covariates, omitting some where personalized allocation based
505   on these covariates seemed unilkely for fairness reasons. The covariates we retain include:
506   length of previous employment, salary, education level, reason for unemployment, region,
507   years of experience at previous job, statistical risk level, job search type (full-time or non-full
508   time), wage target, time of first unemployment spell, job type, and number of children.

509 - Train/validation/test: We train GRF on a 50% training data split and evaluate metrics and on
510   a 50% out of sample split (use the trained GRF model to generate out-of-sample estimates
511   on the test data).

512 - Hyper-parameters: we use GRF defaults for the assessed methods.

513 - Evaluation runs: 50.

514 - Uncertainty quantification: We evaluate the TPRs for 100 percentiles of the rate of CATE
515   estimates over all replications. To compute disparities and ROC curves, we then average
516   the partially identified TPR and FPR at each threshold (e.g. Figs. 1 and 2), and plot the
517   average curve. To simplify discussions we do not quantify uncertainty on the interval itself,
518   noting that since the bounds are closed-form we can circumvent some of the issues regarding
519   inference on LP-based estimators. The definition of *coverage* of inference for interval
520   bounds depends on the parameter of interest (the population parameter or partially identified
521   interval), e.g. see [31].

522 - Computing infrastructure: MacBook Pro, 16gb RAM.

523 An interacted linear model indicates potential heterogeneity of treatment effect with significance on
524 college education, economic layoff, those seeking work due to fixed term contracts or those with
525 previous layoffs; we refer to the original analysis of [11] for additional details.

# C  Substantive Discussion: Fairness vs. Justice

527 We first caveat our use of "disparate impact": while our selection of protected attibutes parallels
528 choices of protected attributes that appear elsewhere in the literature on fair machine learning, for the
529 case of interventions, there may not be precedent from discrimination case law, nonetheless assessing
530 fairness with respect to these social groups may be of concern. We view disparate impact in this
531 domain as assessing fairness of outcome rates under a personalization model.

532 **Should true positive rates be adjusted for?** Our presentation of an identification strategy of
533 fairness metrics for allocating interventions with unknown causal effects begs the question: should
534 disparities in TPR and FNR be adjusted for in the interventional welfare setting? Is responder-accuracy
535 parity a meaningful prescriptive notion of fairness?

Figure 3: Diagnostics for gender protected attribute for Section 7 (not-woman vs. woman)

Figure 4: ROC curves under Assumption 1 for Section 7

One critique of outcome-conditional fair classification metrics recognizes the dependence of false positive rates on the underlying *base rate*, $\mathbb{P}(Y = 1 \mid A = a)$, [20, 21]. The equivalent situation occurs when the within-group ATE varies by the protected attribute, e.g. $\mathbb{E}[\tau \mid A = a]$ differs.

Ultimately, external domain knowledge is required to adjudicate whether group-wide disparities in ATE should be adjusted for, or to decide which normative notion of distributive justice or fairness is appropriate. For example, consider the case of job training. From an economic perspective, multiple mechanisms could explain heterogeneity in CATE by race. Active labor market programs (see [22]) may be less effective for one group vs. another group due to the presence of labor-market discrimination. Alternatively, they could be less effective due to correlation of group status and efficacy that is mediated by occupation choice: one group may be more interested in labor markets where the primary benefits of job search counseling, in reducing search frictions, are not barriers to employment in the first place relative to other factors such as skills gaps. Intuitively, the former mechanism of ATE variation by group reflects a notion of "disparity" which remains problematic, while the latter may seem to reflect an unproblematic causal mechanism. While mediation analysis and fairness defined in terms of path-specific effects could further decompose the treatment effect along these stated mechanisms, in policy settings, collecting all of the relevant information can be burdensome, and deciding on a causal graph can be difficult.

**Claims Across Outcomes** We first outline different frameworks for thinking about fairness/equity of algorithms and interventions. Analogous to the proposals arising from metrics proposed in fairness in machine learning, one might view the decision-maker's concern to be of ensuring *accuracy* parity, that the decisions meted out are overall beneficial to individual. We view a theory of fairness that assesses disparities in outcome-conditional error rates in the context of a theory of normative claims arising from "claims across outcomes". [1] develops a "claims across outcomes" framework of fairness and social welfare, in the context of an overall welfarist theory of justice.

On the one hand, fair classification from the point of view of assessing or equalizing TPR or TNR disparities may be interpreted in a claims context as: for an individual with "true outcome" $Y$ and covariates $X$, an individual with the true label $Y = 1$ as having a comparative claim for $\hat{Y} = 1$, if the predictor $\hat{Y}$ is an allocation tool. We can map the setting of personalized interventions to the "claims across outcomes" setting: the potential outcomes framework posits for each individual the random variables of outcomes $Y(0), Y(1)$. In the responder setting, the true label is responder status $Y(1) > Y(0)$. However, since these are *jointly unobservable*, in situations where heterogeneous treatment effects are plausible, the best guess is an individual-level treatment effect conditional

on covariates, $\mathbb{E}[Y(0) \mid X = x], \mathbb{E}[Y(1) \mid X = x]$. In this interventional setting, one can think
of individuals having claims in favor of favorable outcomes, e.g. a claim in favor of $Y(1)$ if
$Y(1) > Y(0)$.

For the case of interventions, classification decisions $Z$ are allocative of real interventions, and we
argue that implicitly, the consideration of social welfare (balancing efficiency and program costs) is
an important factor in the original design of social programs or personalized interventions. This is
in sharp contrast to the literature on fair classification which considers settings such as lending in
finance, or risk prediction in the criminal justice system, where overriding concerns are primarily
those of *vendor* utility.

On the other side of the spectrum, we can recall axiomatically justified social welfare functions
that apply to the case of *deterministic* resource allocation, where outcomes are generally known. A
decision-maker might also be concerned with equity considerations, adopting a min-max welfare
criterion, appealing to Rawlsian justice frameworks. Another approach is simply assessing the
population cardinal welfare of the allocation, e.g. the policy value $\mathbb{E}[Y(\pi(X))]$ or a social-welfare
transformation thereof, $\mathbb{E}[g(Y(\pi(X)))]$. The literature on policy learning addresses welfare function-
als that are linear functionals of potential outcomes, see [37]. Cardinal welfare constraints such as
those studied in [27] can be applied with an imputed CATE function.

**Comparison to other work on fair classification and welfare.** [41] study the implications of
classifier-based decisions, as well as proposals for statistical parity, on group welfare. Their work
addresses selection rules that have known marginal impacts by group. [29] studies the welfare weights
implied by classification parity metrics and shows that enforcing classification parity metrics are
not Pareto-improving. Rather than studying the welfare implications of classification parity, we are
concerned with assessing non-identifiable model errors in the causal-effect personalized intervention
setting. Since in the personalized intervention setting, welfare is a primary objective for the Planner
(e.g. social services, or social protection more broadly), modulo cost considerations, combining
the distributional information from identification of classification errors with other social welfare
objectives is of possible interest.

We next aim to provide concrete examples of discussions regarding the distributional impacts of
interventions, in order to provide additional context on different contexts wherein different notions of
"fairness" from the fairness in machine learning literature map onto welfare or justice concerns, as
stated in discussions on interventional outcomes.

**Lexicographic fairness or maximin (Rawlsian) fairness.**

In a large multi-site graduation trial on testing an intensive, composite intervention targeted at the
"ultra-poor", which comprised wraparound services including coaching and revenue-generating
resources, still the poorest seemed to benefit least from the intervention in terms of sustained revenue
[7]. In this setting, concerns about maximin fairness (Rawlsian justice) might override considerations
of efficiency insofar as one might be willing to invest resources to help the worst-off on humanitarian
grounds.

**Universalism.**

Criticisms of targeted policies in general note practical difficulties introduced by imposing and
enforcing eligibility guidelines. [48]. Although discussion of resource constraints may be used
to justify a targeting scheme, critics of targeting argue that the most efficient targeting is not as
welfare-improving as simply advocating for greater resources [25].

**Additional distributional preferences on $Y(Z)$ with respect to equitable or redistributive aims
of the policy.**

[14] consider profiling based on covariates as a means of allocating government services, in the
example of allocating predicting unemployment duration to allocate reemployment services. They
outline competing equity vs. efficiency concerns, in the case that unemployment duration is correlated
with treatment efficacy (e.g efficacy of reemployment services), and conclude that " tradeoffs between
alternative social goals in designing profiling systems are likely to be empirically important... the
form and extent of these tradeoffs may depend on empirical relationships between the impacts of
the program being allocated and the equity-related characteristics of potential participants." While
outcome-conditional true positive rates or true negative rates compare model performance across
binary protected attributes, program designers may remain concerned regarding the distribution of

benefits. [17] consider "removing the veil of ignorance" under the simplifying of constant treatment response to consider distributional (quantile) treatment effects, as a relaxation of the anonymity axiom of cardinal social welfare. Distributional preferences are relevant when program designers are concerned about model performance at finer-grained levels than discrete protected attribute.