[Reviews · NeurIPS 2019]

Reviewer 1



This paper introduces metrics for quantifying the fairness of a personalized intervention (with binary treatment & outcome) that are analogous to fpr/tpr in the classification setting. They show how these metrics are identifiable under the assumptions of strong-ignorability and monotone treatment response (treatment is never harmful). Developing appropriate measures of fairness for personalised interventions is an important questions, with significant potential application and this paper is well structured and clearly written. The identification results themselves are not particularly novel as similar results have been proven in other settings, but the context is new. I have read the author response and other reviewers comments and maintain that this paper is worth accepting because: ML systems recommending personalised interventions in sensitive settings, such as improving educational results or reducing recidivism are widely encountered in practise in areas where we are concerned with fairness. So this paper represents an important attempt to extend existing fairness metrics to this more complex setting. While I think it the appropriateness of quantifying fairness in terms of metrics that are not identifiable without monotonicity assumptions (even with unlimited randomised experimental data) remains up for debate, this is a debate that is worth having within the NeurIPS community. This paper is well written and argued and represents a good starting point for that debate.

Reviewer 2



While the basic ideas explored here are interesting, I have several concerns with this work: (1) The primary motivation or contributions of the paper are not well articulated. The title and writing at various places suggest "personalized interventions". But, the methods for auditing policies for fairness as well as the evaluation results are centered around scenarios where interventions are conducted in a randomized and controlled (i.e., non-personalized) way. So in what way is the work related to personalized interventions, beyond perhaps, making a case for personalized interventions to achieve fairness? (2) The focus of much of the main (8-page) paper is on unfairness measures (disparity in TPR and TNR) and how to estimate them, but not on mechanisms to achieve them. This makes me wonder how well suited the paper is to NeurIPS and if this paper would not be a better fit for a conference focussed more on fairness. Because one might argue that, while the ideas are conceptually interesting, technically the paper does not contribute much. (3) I think there is an error on line 168 -- it is hard to distinguish between p_{10} and p_{00} rather than p_{10} and p_{11}. Can you check this again?

Reviewer 3



The paper considers an important and critical societal matter and is well-written. Personalized interventions are ideal in cases where everyone has access to the same resources, but resources are limited and intervention on one’s treatment may impact interventions on other individuals in the population. In other words, decisions may interfere with each other. This has been studied in this recent paper: https://icml.cc/Conferences/2019/ScheduleMultitrack?event=4860. Also, fair personalized intervention has been considered in this paper: https://icml.cc/Conferences/2019/ScheduleMultitrack?event=5183 where the setting is very similar to the one being considered in this paper (offline optimal policy learning under fairness constraints). I was wondering what the authors’ thoughts are on these two papers? and how limited resources impact their disparate impact metrics?

Reviewer 4



Update: I have read the other reviewers' comments and the author feedback, which has adequately addressed my concerns. My overall score remains unchanged. ============ Originality. Proposition 2 seems to be the main novel observation of this work. Although it is straightforward, it is nice to point out that the monotone treatment response assumption, which is reasonable for 'positive'/beneficial interventions, allows for identification in this case. The insight seems to be present in previous work [32], so extending it to give expressions for TPR/FPR is not particularly original though still useful. The partial identification results are quite novel, though I was not sure of the practicality. How should uncertainty sets for p_10 be chosen? Is there any way to estimate how much the monotone treatment response assumption is violated in practice? Given the monotone treatment response assumption, many "non-causal" results from previous work can be applied without modification, e.g. [34]. Robust ROC and xROC extends the standard tools to the partial identification case, but it's not clear how useful these notions are/if they provide any additional insight, even in the case study provided. They seem to be too pessimistic and it's hard to interpret in the curves in Figure 2. Quality. Paper was well written. Clarity. Writing and results were clear. Figure 2 had too many curves and it was unclear how to interpret. Significance. It is important to address identifiability issues for 'fairness metrics' in applications involving interventions (and therefore missing data). Empirical verification was somewhat limited, since only 1 dataset was used.

[Author Response · NeurIPS 2019]

**Reviewer 1**   Re: suggested improvements: Section 2: Thank you for the reference and note. We will add the citation and discussion. That is *exactly* why we provide sensitivity analysis alongside the monotonicity assumption; in this context, it attends to "Moral 2" of Dawid (2000) by changing the goal to set- rather than point-identification, under varying strengths of additional assumptions, which may be plausible in practice.

Re: ignorability: we actually only need weak ignorability; thanks for catching. Section 3: Thank you for the references; we will add to the related work section on partial identification, alongside the Balke and Pearl ref. To clarify, the main point of departure from previous work is in addressing non-identifiability when **conditioning on the counterfactual potential outcome** and in **providing bounds for non-linear functionals**. Re: dependencies: We will add: our code uses numpy/sklearn/pandas, etc. We use the R Generalized Random Forests package for causal effect estimates.

**Reviewer 2**   1) We disagree. Our introduction cites **many** works that learn CATE (personalized causal effect) and personalized interventions from RCT (or, observational data); e.g. [17,23] for homelessness prevention and job training interventions. To clarify: these personalization approaches learn CATE and policies in "batch" rather than online fashion. The aim is still personalization; but the batch data *must* necessarily involve some randomization/overlap/exploration. When assessing the potential impact of a personalized policy, we show that this causal setting poses identification challenges for fairness metrics and provide estimators and sensitivity analyses.

2) Firstly, we do provide means of adjustment via Hardt et al. [26]. But we do highlight that direct adjustment of group-specific thresholds is controversial in practice and its relevance context-dependent, and this is not limited to our setting. We therefore defer the substantive (and less technically contributory) discussion to the appendix. In the appendix, we extensively discuss alternative approaches for minimizing disparities, including adjustment and covariate choice. Because TPR/FPR disparities could arise for a variety of reasons, it is not clear that adjustment of predictions is necessarily beneficial; we discuss reasons for caution in the appendix.

3) There is no typo there. Thanks for checking!

**Reviewer 3:**   "I was wondering what the authors' thoughts are on these two papers ..." Thank you, we will include these two references and discussion. There are different types of interference: 1. A universal budget/resource constraint; 2. Operational constraints (e.g. assignment under unit capacity constraints), 3. Network-type interference (violations of SUTVA) such as peer effects, and 4. general-equilibrium interference. 1&2 are related to resources. In the case of 1, under a universal budget, the optimal policy is to treat everyone above some quantile of CATE (e.g. [15]). This is an important motivation for our approach, since realistic budget constraints would lead to optimal decision policies which threshold CATE; we will highlight this further. Re: 2: Instead of taking Z to be a threshold on CATE, our approach also applies to assessing TPR/FPR of any policy Z, which may optimize assignment under more complicated resource constraints. 3&4 are types of interference that we do not address, we focus on assignment under heterogeneous effects under SUTVA.

Re: Nabi et al 2019: Their approach is complementary. While they adjust for fairness via constrained estimation (constraining pre-specified path-specific effects), they assess policy value via utility that marginalizes over the individuals' labels (essentially utility-weighted accuracy). *If* their approach sought to *also* compare the analogous TPR and FPR (e.g. whether the disutility of fair policies falls on actually-guilty or actually-innocent), they too would have the issue of non-identifiability that we study and address in our work. Similarly with Kusner et al. 2019: their parity constraints are resource equity constraints, not classification parity, conditional on potential outcomes under assignment.

**Reviewer 4**   Re: choosing uncertainty sets: The magnitude of $B$ can be directly calibrated against ATE effect size estimates from similar interventions, mechanistic knowledge, negative controls, or prior distributions on effect sizes, which practitioners typically can reason about. But instead of choosing a single $B$, usually sensitivity analysis is viewed as determining how big a violation is needed to overturn a conclusion. For example, it is unlikely that job training causes someone to not get a job, so if we need $B \geq 0.05$ to overturn a conclusion then it is robust if it is unrealistic 5% of the population would experience a negative causal effect. Re: estimating level of violation: Unfortunately, the level of violation is itself also unidentifiable without additional data like negative controls (see above). Re more datasets: There are not many *publicly available* datasets that were both large enough to reasonably support learning CATE as well as out-of-sample evaluation, had convincing protected group info, binary outcomes, *and* plausible monotonicity. That is why we introduced the Behaghel et al dataset, which we think is an exciting new dataset for considering fairness.

Re: Robust ROC and xROC: These are intended to provide additional information, in analogy to the use of ROC curves in assessing risk scores. Since sensitivity analysis focuses on illustrating how the extent of various claims (here, possible conditional disparities in performance) changes with the varying violations of assumptions (defier probability), we show how the bounds loosen with increasing violation. That some bounds are large should caution an analyst to draw hasty conclusions, while tight bounds imply a robust conclusion: our case study includes both examples. In the main text, the curves are overlaid: we will break out these as individual figures in the appendix and explain further how an analyst should interpret regions of overlap or non-overlap of these curves.

[Meta-Review · NeurIPS 2019]

The reviewers liked the focus on personalized interventions and the effort to reframe the discussions of fairness around the more challenging issue of inferring treatment outcomes for individuals rather than in aggregate. Their approach is well thought out and well developed. The reviewers were more than satisfied with the author response.